# Empirical Training Time Prediction for LLM Fine-Tuning Using Scaling Laws

Chianing Wang
Toyota InfoTech Lab USA & Santa Clara University
johnny.wang@toyota.com

Younghyun Cho
Santa Clara University
younghyun.cho@scu.edu

*Abstract*—This paper presents a methodology to efficiently estimate the training time and associated computational cost of fine-tuning Large Language Models (LLMs). Our approach introduces a novel two-stage methodology that incorporates an intelligent tuning algorithm called the Scaling Laws Smart Tuning (SLST) algorithm for efficient sample data collection and a time prediction model combining scaling laws with Gradient Boosting techniques. The scaling laws capture broad training trends concerning model parameters and dataset sizes, while Gradient Boosting models effectively reduce residual errors by modeling complex nonlinear relationships directly from data. Through this integrated approach, we achieve a high accuracy in training time predictions, which can significantly enhance resource planning and infrastructure decision-making. Our results demonstrate the effectiveness of the methodology, which balances interpretability and predictive accuracy, and highlight its scalability across various computational and parallelism environments.

## I. BACKGROUND AND RESEARCH QUESTION

### A. Infrastructure Cost Estimation for LLM

The rapid advancement of generative AI and Large Language Models (LLMs) [2] has transformed numerous industries. Foundation models [1], typically large-scale models pre-trained on extensive datasets, capture general knowledge and can be fine-tuned for various downstream tasks. Despite their success, substantial computational and financial demands during pre-training and fine-tuning [5] present significant challenges. As investments in LLMs grow, accurately estimating infrastructure costs becomes critical for efficient resource management, evaluating project feasibility, and guiding strategic infrastructure decisions.

### B. Scaling Laws for LLM

The LLM scaling laws by Kaplan et al. [13] empirically quantify the relationships among model size ($N$), dataset size ($D$), and computational resources required ($C$). Pioneering studies on models like GPT-3 [2] demonstrated that power-law relationships effectively model training-related costs. Initially drawn from physics and natural sciences, these scaling laws characterize nonlinear and predictable scaling behaviors. However, much existing research has primarily focused on predicting training loss rather than training time, creating a noticeable gap in the fine-tuning context.

### C. Improving LLM Fine-Tuning Performance

While large-scale pre-training remains resource-intensive and primarily accessible to well-resourced entities, fine-tuning has become increasingly feasible through Parameter-Efficient Fine-Tuning (PEFT) techniques [6], particularly Low-Rank Adaptation (LoRA) [11]. LoRA selectively updates a small subset of model parameters, significantly reducing computational overhead. It operates independently from parallel computation methods such as Pipeline Parallelism (PP) [12] and Tensor Parallelism (TP) [17], which primarily manage computational distribution rather than directly influencing learning dynamics.

### D. Motivation and Research Question

This work proposes a methodology for efficiently and accurately estimating fine-tuning training times for LLMs. While significant research exists on LLM training dynamics, most studies emphasize training loss rather than training time, leaving several key research gaps:

1) **Emphasis on Pre-training Over Fine-tuning:** Most studies have primarily examined scaling laws within pre-training scenarios. For instance, Hernandez et al. [10] conducted an extensive analysis of scaling behaviors for model transfer and pre-training, offering valuable insights into optimal model sizes and resource allocation. While their findings could potentially be adapted to fine-tuning contexts, their primary focus remains pre-training, which inherently differs in runtime characteristics compared to fine-tuning. Additionally, although predictions based on pre-training data might theoretically apply to fine-tuning scenarios, practical experimental validation remains lacking, leaving the feasibility of such adaptation uncertain.

2) **Focus on Training Loss Instead of Runtime Prediction:** Existing research typically emphasizes predicting and minimizing training loss, often neglecting explicit modeling of runtime, which is crucial for practical resource management and scheduling. Zhang et al. [22] provided comprehensive runtime analyses across various stages of LLM development, yet generalized predictive frameworks remain sparse.

3) **Limited Integration of Scaling Laws with Empirical Fine-tuning Time Prediction:** Scaling laws have been foundational in understanding pre-training performance, but integration into runtime prediction methodologies for fine-tuning remains relatively unexplored. Xia et al. [21] developed analytical methods for estimating fine-tuning

costs on specific GPU setups, but their approach does not explicitly employ scaling laws for generalized prediction across different computational environments.

Addressing these gaps, this paper aims to answer the following research question: *How can power-law scaling relationships effectively predict computational costs (training time) for LLM fine-tuning, thereby enabling efficient and sustainable AI infrastructure planning?*

## II. Two-Stage Methodology

To address the aforementioned research question, we propose a comprehensive approach combining theoretical modeling and empirical validation. Our method involves two primary stages: first, efficient sample data collection guided by the Scaling Laws Smart Tuning (SLST) algorithm to ensure data quality and adherence to the LLM Scaling Laws; second, leveraging the collected data to predict training time ($T$) using redesigned LLM Scaling Laws formulas. Constants and exponents within these formulas are determined via the curve-fitting method, and the residuals, which are the differences between observed and predicted training time, are further minimized using Gradient Boosting.

### A. Scaling Laws Smart Tuning

Our method begins with sample collection via fine-tuning LLMs. To ensure valid and predictive training time ($T$) data, the training loss ($L$) must conform to LLM Scaling Laws. Concurrently, hyperparameters such as learning rate and batch size are tuned during sample data collection, explicitly guided by the Scaling Laws. The LLM Scaling Laws provide a reliable and interpretable framework, characterizing the relationships among model size ($N$), dataset size ($D$), and training dynamics. By leveraging it, we can significantly reduce the number of empirical trials while ensuring the quality of the collected data samples, that is suitable for training time predictions. To formalize this approach, we introduce the Scaling Laws Smart Tuning (SLST) algorithm (Algorithm 1). The algorithm operates as follows:

- Pre-defines the key parameters such as model sizes $N[\,]$, maximum dataset size $D$, e.g., 80k # of tokens (22MB), and dataset intervals (lines 1–3).
- Initializes the hyperparameters such as *learning rate* ($lr$) and *batch size* ($bs$) along with their bounds (lines 5–9).
- Trains the language model using the $llm\_fine\_tuning()$ function with current parameters (line 11).
- Evaluates training logs at specified intervals(lines 12–13), as illustrated in Figure 1, ensuring the loss $L$ adheres to the LLM Scaling Laws which is the $L$ should improve as we increase $N$ and $D$ (lines 14–19).
- If non-compliance occurs, adjusts the hyperparameters and retrains (lines 20–26).
- Outputs optimal parameters upon compliance and progresses systematically through dataset intervals, as demonstrated in Figure 2, then advances to subsequent model sizes (lines 28–31).

---

**Algorithm 1** Scaling Laws Smart Tuning Algorithm

---

**Input:** Define ML models $N$, max data sizes $D$ with interval $range$.
Define initial batch size $bs$ and learning rate $lr$.
Define batch size, learning rate boundary ($min\_bs, max\_lr$).
**Output:** Optimal learning rate $lr$, batch size $bs$, loss $L$, time $T$

1: $N[\,] \leftarrow get\_models\_list()$
2: $D \leftarrow get\_max\_dataset()$
3: $range \leftarrow get\_dataset\_range(D)$
4: **for** each n in $N[125m, 1.3b, \ldots, 66b\}$ **do**
5:    $lr \leftarrow get\_initial\_lr\_by\_model(n)$
6:    $bs \leftarrow get\_initial\_bs\_by\_model(n)$
7:    $max\_lr \leftarrow get\_max\_ls\_by\_model(n)$
8:    $min\_bs \leftarrow get\_min\_bs\_by\_model(n)$
9:    Initialize parameters: $totalTime \leftarrow 0$, $prevDLoss \leftarrow 0$, $currD \leftarrow 1$
10:    **while** $currD \neq patterns[range]$ **do**
11:      $log \leftarrow llm\_fine\_tuning(lr, bs, n, D)$
12:      $step \leftarrow get\_log\_step(log)$
13:      $patterns[\,] \leftarrow np.linspace(1, steps, range)$
14:      **for** each d in $patterns[currD, \ldots]$ **do**
15:        read $log$ file at $d$
16:        extract values $Loss, Time$ from $log$ file
17:        $prevNLoss \leftarrow get\_previous\_loss(N[n-1], d)$
18:        $currLoss \leftarrow Loss$
19:        **if** $currLoss > prevNLoss$ **or** $currLoss > prevDLoss$ **then**
20:          **if** $bs > min\_bs$ **then**
21:            $bs \leftarrow bs/2$
22:          **else if** $lr < max\_ls$ **then**
23:            $lr \leftarrow lr \times 2$
24:          **end if**
25:          $currD \leftarrow d$
26:          **break**
27:        **else**
28:          $totalTime \leftarrow totalTime + Time$
29:          **Output:** $lr$, $bs$, $currLoss$ as $L$, and $totalTime$ as $T$
30:          $totalTime \leftarrow 0$
31:          $prevDLoss \leftarrow currLoss$
32:        **end if**
33:      **end for**
34:    **end while**
35: **end for**

---

### B. Training Time Prediction

The second stage consists of two phases: Scaling Law $T$ formulation and Gradient Boosting.

*1) Scaling Laws T Formulation:* Leveraging SLST, we collect comprehensive sample data across various GPU and parallelism configurations. We analyze this sample data using SciPy's optimization curve fitting module [18], employing nonlinear least squares to fit the redesigned training time scaling laws formula and calibrate our own formula, effectively capturing the inherent nonlinearities.

Power laws [20] characterize nonlinear relationships as straight lines on log-log plots, forming the basis for LLM Scaling Laws. While LLM Scaling Laws for training loss ($L$) typically show decreasing trends by dividing model size ($N$) and dataset size ($D$) by constants, training time ($T$) shows increasing trends. First, to predict training time in regards with $N$ and $D$, we adapt a scaling formula by multiplying $N$ and $D$

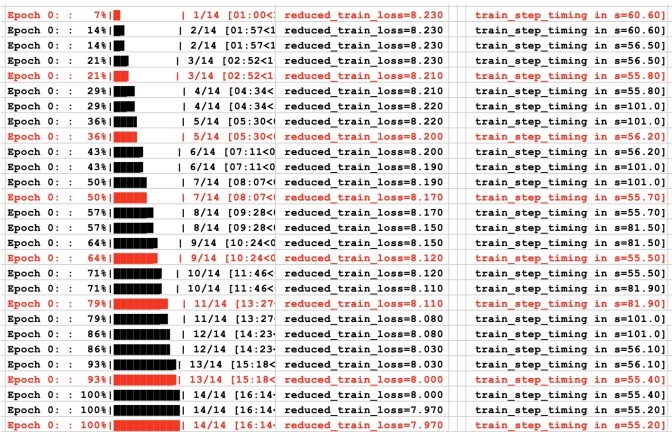

**Fig. 1:** SLST Log Example

Model Size N=125m and Dataset Size D=80k with 14 steps and 8 ranges(intervals)

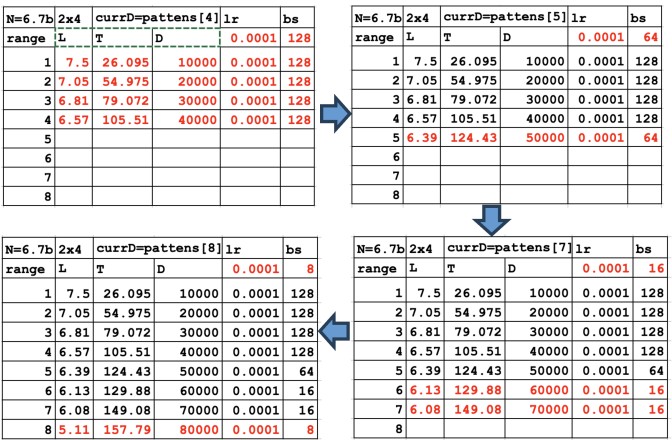

**Fig. 2:** SLST trys diff lr and bs with Result Example

Model Size N=6.7b, G (# of GPUs)=8 and PPxTP=2x4; L is Training Loss, T is Training Time, D is Dataset Size (# of Tokens), lr is Learning Rate and bs is Batch Size

with constants, aligning with the observed increasing behavior in formula (1).

$$T(N, D) = \left[ \left( N_c N^{\frac{\alpha N}{\alpha D}} \right) + D_c D \right]^{\alpha D} \tag{1}$$

Then, in order to model the training time for Pipeline Parallelism (PP) and Tensor Parallelism (TP), each of which would contribute to reduce the training time, we extend formula (1) with the TP and PP time factors that are modeled via two approaches: first, Power law based TP (2) and PP (3) factors, and second, Amdahl's law based TP (4) and PP (5) factors, as follows.

$$Factor(TP)_P = \left( \frac{TP_c}{TP} \right)^{\alpha TP} \tag{2}$$

$$Factor(PP)_P = \left( \frac{PP_c}{PP} \right)^{\alpha PP} \tag{3}$$

Amdahl's Law [9] illustrates performance limits in parallelization contexts. It highlights diminishing returns as parallel

| Formula | Base and Factors |
|---|---|
| $T_A(N, D, TP, PP)$ | $\left[ \left( N_c N^{\frac{\alpha N}{\alpha D}} \right) + D_c D \right]^{\alpha D} \times \left[ \left( \frac{\gamma}{TP} + 1 - \gamma \right) \right] \times \left[ \left( \frac{\delta}{PP} + 1 - \delta \right) \right]$ |
| $T_P(N, D, TP, PP)$ | $\left[ \left( N_c N^{\frac{\alpha N}{\alpha D}} \right) + D_c D \right]^{\alpha D} \times \left( \frac{TP_c}{TP} \right)^{\alpha TP} \times \left( \frac{PP_c}{PP} \right)^{\alpha PP}$ |

resources grow, providing accurate training-time improvement estimates:

$$Factor(TP)_A = \left[ \left( \frac{\gamma}{TP} + 1 - \gamma \right) \right] \tag{4}$$

$$Factor(PP)_A = \left[ \left( \frac{\delta}{PP} + 1 - \delta \right) \right] \tag{5}$$

Note that the factors $Factor(PP)$ and $Factor(TP)$, which represent the scaling behavior due to GPU, tensor, and pipeline parallelism, respectively, are explicitly combined into single, comprehensive equations. For clarity and concise reference, these unified equations, including base terms and both factors, are summarized in Table I:

*2) Residual Reduction via Gradient Boosting:* Gradient Boosting [8], a sequential ensemble machine learning technique, minimizes pseudo-residuals iteratively. It enhances the Training Time Scaling Laws through a hybrid approach: initially fitting a physics-inspired scaling model to broadly capture trends, then employing Gradient Boosting (e.g., XGBoost [3], LightGBM [14], or CatBoost [16]) to refine predictions by learning residual nuances. This combination effectively integrates interpretability with predictive accuracy, significantly enhancing model reliability.

Gradient Boosting [8] enhances the Scaling Laws model by minimizing residuals through a sequential ensemble of weak learners. This approach refines predictions beyond the Scaling Laws' broad trends, significantly improving accuracy and robustness.

## III. EXPERIMENT

To validate our time prediction methodology, we conducted comprehensive experiments utilizing the Meta OPT (Open Pretrained Transformer) model [23], selected due to its status as a robust foundation model and the extensive range of model sizes available. In our experiments, we focus on fine-tuning LLMs using the Low-Rank Adaptation (LoRA) [11] technique. The dataset selected for our experiments is the Self-Instruct dataset [19], containing 52k instructions along with 82k paired inputs and outputs. The dataset was segmented into eight intervals, ranging from 10,000 to 80,000 tokens, and sizes from 2.6MB to 22MB. We utilized a computing cluster equipped with 16 NVIDIA H100 GPUs [4] and leveraged the Megatron framework [17], enabling various experimental scenarios involving different GPU counts and levels of pipeline and tensor parallelism.

### A. Scaling Laws Smart Tuning

We first compare our SLST algorithm against a simple grid search approach. The grid search exhaustively explores 16 scenarios for each model-dataset combination, using 4

**TABLE II:** Number of Trials: Brute Force Grid Search vs. SLST

| D(=80K) \ N | 125M | 1.3B | 2.7B | 6.7B | 13B | 30B | 66B | **Total** |
|---|---|---|---|---|---|---|---|---|
| SLST Trials (P=4) | 1 | 1 | 1 | 1 | 2 | 2 | 4 | **12** |
| Brute Force (P=4) | 128 | 128 | 128 | 128 | 128 | 128 | 128 | **896** |
| SLST Trials (P=8) | 0 | 1 | 1 | 1 | 3 | 1 | 4 | **11** |
| Brute Force (P=8) | 0 | 128 | 128 | 128 | 128 | 128 | 128 | **768** |
| SLST Trials (P=16) | 0 | 0 | 1 | 1 | 0 | 3 | 4 | **9** |
| Brute Force (P=16) | 0 | 0 | 128 | 128 | 0 | 128 | 128 | **512** |

**TABLE III:** Test Results for Self-Instruct Dataset

| Formula | MAE_S ($\downarrow$) | $R^2$_S ($\uparrow$) | MAE_C ($\downarrow$) | RMAE_C ($\downarrow$) | $R^2$_C ($\uparrow$) |
|---|---|---|---|---|---|
| $T_A()$ | **20.974** | **0.790546** | **12.710** | **16.695** | **0.918035** |
| $T_P()$ | 20.963 | 0.790435 | 13.307 | 17.624 | 0.908659 |

Testing Dataset D=60k (16MB)~80K (22MB) with Masking Out non-compliance LLM Scaling Laws results for different # of GPU and Parallelism Configurations. The metrics with _S means performance result after 1st phase: scaling laws $T$; the metrics with _C means performance result after 2nd phase: Gradient Boosting (CatBoost).

different learning rates and 4 different batch sizes. Partial experimental results for configurations with $PP \times TP = 16 \times 1$, $PP \times TP = 8 \times 2$, and $PP \times TP = 4 \times 4$ using 16 GPUs are presented in Table II. The results clearly illustrate SLST's efficiency in significantly reducing the number of trials required during sample data collection.

*B. Training Time Prediction*

Following the sample data collection via SLST, we evaluated two candidate scaling laws formulas, which are Amdahl's law-based $T_A()$ and Power Law-based $T_P()$, presented in Table I. Sample data collected using SLST from 10k (2.6MB) to 50k (13MB) tokens served as the training set, while the 60k (16MB) to 80k (22MB) tokens range was reserved for testing. The GPU configurations varied systematically, with the total number of GPUs ($G$) ranging from 16 down to 8, and the pipeline parallelism ($P$) adjusted accordingly—for instance, when $G = 16$, configurations included $PP \times TP = 16 \times 1$, $8 \times 2$, and $4 \times 4$, among others, ensuring comprehensive coverage of parallelism combinations. Constants and exponents within the formulas were determined using curve fitting, and residual errors were further minimized using CatBoost. Test results comparing the models are shown in Table III.

The results in Table III indicate the superior performance of the $T_A()$ model, evidenced by the lowest MAE and RMAE and the highest $R^2$ scores. These outcomes highlight the effectiveness of Amdahl's law-based factors in modeling GPU and parallelism scaling more accurately than the power law-based factors. Optimized constants and exponents example for Self-Instruct Dataset for the $T_A()$ model after curve fitting are listed in Table IV. Further examples, shown in Figures 3 illustrate the prediction trends clearly across various test scenarios, demonstrating the scalability of model sizes and dataset intervals.

## IV. CONCLUSION AND FUTURE WORKS

In conclusion, this research presents a robust methodology for accurately predicting training time and computational costs associated with LLM fine-tuning. We introduced a two-stage predictive framework, incorporating the novel Scaling Laws Smart Tuning (SLST) algorithm for efficient sample data

**TABLE IV:** Constants and Exponents Example for Self-Instruct Dataset

| **Constant and Exponents for $T_A()$** | **Values** |
|---|---|
| $N_c$ | 9.52074184e-04 |
| $\alpha N$ | 7.92198609e-01 |
| $D_c$ | 1.72112915e-044 |
| $\alpha D$ | 2.17499781e+00 |
| $\gamma$ | 6.52994195e-30 |
| $\delta$ | 9.80222336e-01 |

collection alongside a time predictive approach. This time prediction model combines the LLM scaling laws with time modeling for GPU parallelism, and we use Gradient Boosting to enhance prediction accuracy. The approach can effectively capture broad trends related to model parameters and dataset sizes, while Gradient Boosting models further refine these predictions by modeling nonlinear relationships and reducing residual errors. This integrated approach achieves high accuracy and reliability, striking a balance between explainability and predictive flexibility, which can be significantly helpful in improving resource planning and informing strategic infrastructure decisions across diverse computational and parallelism environments.

For future work, we plan to further enhance the robustness and broader applicability of our approach by:

- Expanding our experimental dataset with larger sizes and various types of datasets, to improve the generalizability and robustness of predictions.
- Evaluating the methodology's applicability advances not only OPT-based fine-tuning but also extends to other LLMs such as Llama [7] [15], future pre-training phases, inference tasks, and potentially diverse domains like Computer Vision.
- Investigating the effectiveness of our methodology across various GPU vendors and extending this to include infrastructure components beyond GPU, such as networking, storage, power, and thermal management.

While initial sampling and the two-phase prediction approach remain essential for calibrating predictive models, subsequent predictions significantly reduce the need for exhaustive empirical evaluations. By enabling accurate estimations of training times and computational resources across various configurations, our predictive methodology optimizes resource allocation, reduces operational costs, and supports effective strategic decision-making, particularly in large-scale or unexplored scenarios. These initiatives will further demonstrate the versatility and extend the practical utility of our predictive model, ensuring comprehensive adaptability and robust performance across diverse computational environments.

## V. ACKNOWLEDGEMENTS

This work was supported by Toyota Motor Corporation and Toyota InfoTech Labs USA. We used the GPU computing resources and the high-performance server infrastructure at Toyota InfoTech Labs USA for the experiments of this work. We thank Ryokichi Onishi, Hiroshi Abe, and Onur Altintas for their comments and feedback on this work. We also thank Purvang Lapsiwala for his technical support.

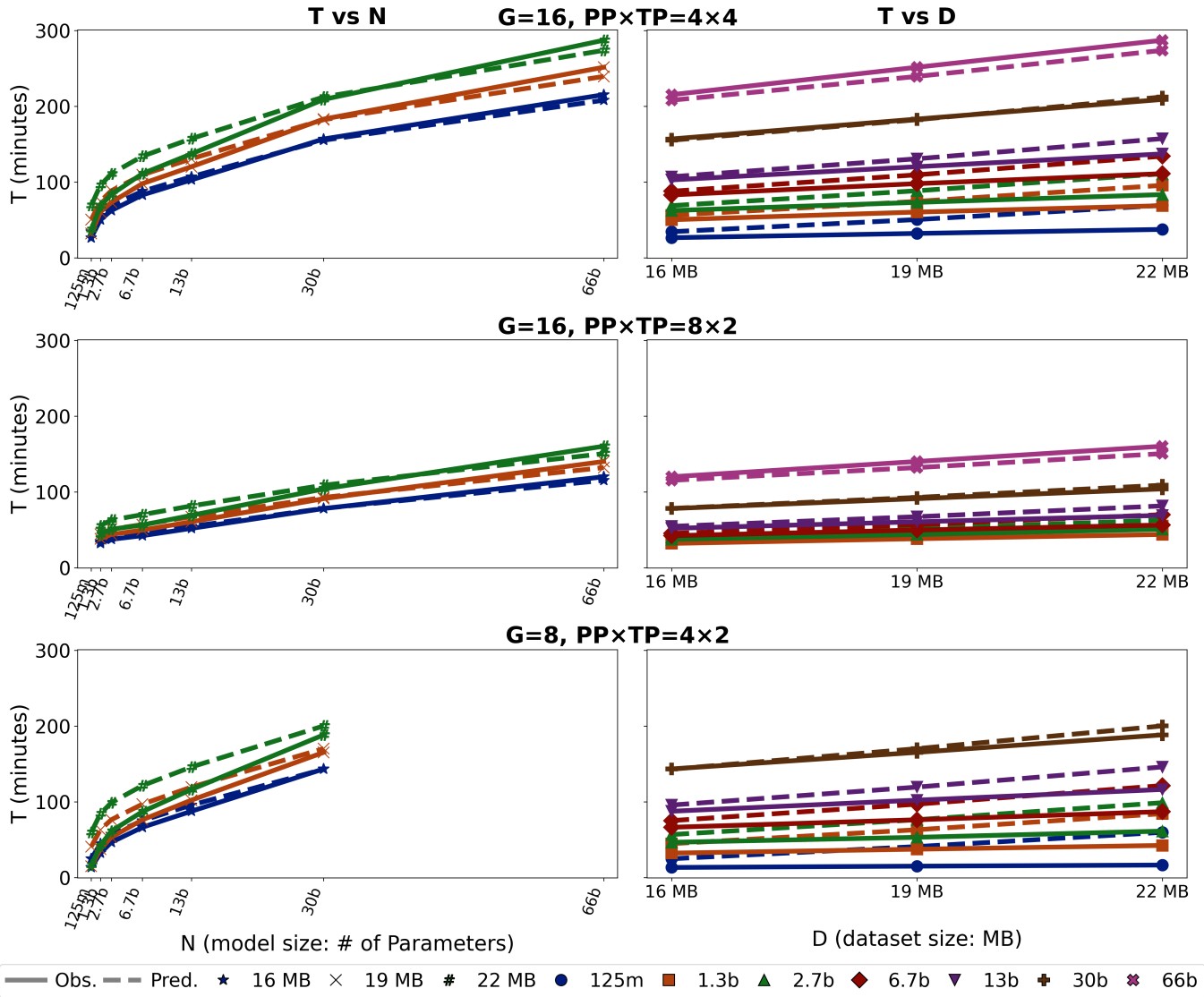

**Fig. 3:** Training time vs model size (left) and Self-Instruct dataset size (right) across G/P settings prediction examples. Our experiments cover model sizes ranging from $N = 125m$ to 66b parameters, with testing dataset sizes between $D = 60K$ tokens (16MB) and 80K tokens (22MB). We present example prediction results for various GPU parallelism configurations, denoted as $(G, PP \times TP)$, specifically $(16, 4 \times 4)$, $(16, 8 \times 2)$, and $(8, 4 \times 2)$. The 8-GPU setup supports fine-tuning only up to the 30B parameter model. Additionally, the configurations $(16, 8 \times 2)$ and $(8, 4 \times 2)$ cannot accommodate all tested model sizes because certain model architectures have numbers of layers or attention heads that are not divisible by the specified pipeline or tensor parallelism factors.

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
