# OpenReview forum: "Empirical Training Time Prediction for LLM Fine-Tuning Using Scaling Laws"
_iscaconf.org/ISCA/2025/Workshop/MLArchSys — MLArchSys 2025 Oral_

### Official Review · Reviewer_Qw28 · 2025-05-16
**Practical Efforts in Fine-Tuning with Scaling Laws, but Limited Novelty and Generalization**

**Confidence:** 3
**Rating:** 5

**Detailed Feedback And Questions For Authors:**

The paper targets a practical and relevant problem in ML system design—predicting the training time of LLM fine-tuning runs—but the proposed solution is limited in both novelty and depth. The combination of power-law fitting and gradient boosting is a standard hybrid modeling strategy, commonly used in performance prediction. The introduced SLST algorithm is essentially a heuristic-guided hyperparameter sweep with scaling-law compliance checks, which, while efficient, is algorithmically straightforward and not conceptually novel.

The paper also lacks sufficient theoretical justification for its core modeling assumptions. The proposed formulas for training time scale as additive power-law functions of model and data size, with GPU parallelism handled via separable multiplicative factors inspired by Amdahl’s law. However, these functional forms are not derived from first principles or any analytical approximation of real training behavior. There is no discussion of their limitations, sensitivity, or how closely they track empirical hardware behavior beyond curve fitting.

Furthermore, the integration of gradient boosting (e.g., CatBoost) is treated as a black-box residual corrector, without analysis of what additional patterns it captures or how it generalizes. The boosting stage lacks transparency—there is no insight into feature importance, error modes, or overfitting risk. This weakens the credibility of the “hybrid” framework, as it’s unclear how much of the predictive power stems from the scaling law itself versus pure data-driven learning.

The evaluation is also limited in scope. All experiments are conducted on the OPT model family using LoRA on a single dataset, with no validation on other model architectures or tuning methods. Without broader testing, it’s difficult to judge the generality or robustness of the method. Additionally, comparisons to stronger baselines (e.g., Bayesian tuning, analytical estimators) are missing, and the reproducibility of the framework is hampered by the lack of open code or config-level details.

In summary, while the motivation is strong and the system setup is cleanly described, the paper’s contributions are mostly empirical and incremental. It would benefit from deeper analysis, broader validation, and more rigorous treatment of its modeling assumptions.

**Top Reasons To Accept The Paper:**

The paper tackles an important and practical problem: predicting LLM fine-tuning time across parallel hardware configurations.

The proposed hybrid methodology, combining scaling laws with gradient boosting, is well-motivated and achieves promising empirical accuracy.

**Top Reasons To Reject The Paper:**

The technical novelty is limited; the scaling laws and boosting combination is straightforward and largely heuristic in implementation.

Evaluation is restricted to a single model family (OPT) and one dataset, limiting the generalizability of findings.

Key design choices (e.g., feature selection, hyperparameter ranges, fitting strategy) are underexplained or lack ablation.

---

### Official Review · Reviewer_iALD · 2025-05-18
**The paper provides significant gains in accurately predicting fine-tuning training time, albeit the writing of the paper could be stronger to make it easier for readers to understand the intuition of the work and takeaways from the results.**

**Confidence:** 3
**Rating:** 6

**Detailed Feedback And Questions For Authors:**

Thank you for submitting this work to MLArchSys'25! I found this paper quite intriguing and enjoyed reading it. Below are a few questions and feedback that would strength this work.

1. The paper notes that previous work focuses on pre-training, not fine-tuning. Further discussion on why this work cannot apply to the fine-tuning environment would be helpful to contextualize the novelty of this work. Simply stating that these works focus on pre-training is not enough to claim why they are insufficient.

2. The results of the work are pretty good, but the explanation throughout the writing could be better. It took several reads and careful parsing of the table and figures to extract the takeaways. Distilling the takeaways in the writing would be very helpful for readers.

3. I'm confused about the use case of this work. For a given model to fine-tune, do we first need to obtain some samples via fine-tuning to then build a training set for fitting the hyperparameters of the candidate scaling law formulas? If so, what is the benefit of making this prediction at all? Ideally, we would know the fine-tuning time before performing any fine-tuning jobs.

4. Can the same hyperparameters of the scaling law formulas work across LLMs? Or are they LLM-specific? If it's the latter, then question (3) above becomes even more prevalent. What is the benefit of making this prediction, given that we require obtaining training data per LLM?

**Top Reasons To Accept The Paper:**

The paper presents a novel comprehensive algorithm to tune scaling law algorithms that accurately predict the optimal batch, learning rate, training loss, and training time for fine-tuning jobs. The algorithm shows significant initial gains in accuracy with time prediction (Table III and Figure 3).

**Top Reasons To Reject The Paper:**

The writing throughout the paper could be stronger: (1) better explain the proposed new scaling laws and algorithm, and (2) distill the takeaways from the figures and tables.

---

### Official Review · Reviewer_R7gi · 2025-05-18
**Review summary**

**Confidence:** 5
**Rating:** 5

**Detailed Feedback And Questions For Authors:**

This paper presents a method for estimating the training time of Large Language Models (LLMs) fine-tuning with LoRA.

- Pros
1. The problem is well defined and the gaps mentioned in section1 provide clear motivations for this work
2. Experimental results validate the effectiveness of the proposed method

- Cons
1. The proposed method relies on loss curves for quality metrics, thus it is unclear how the proposed design tackle fine-tuning without clear loss reduction signals.
2. The title of this work appears to overclaim as LoRA is the only fine-tuning method tested.
3. This work considers PP (pipeline parallelism) and TP (tensor Parallelism) as the efficiency factors. How to incorporate additional compute diagrams (other sharding or parallelism configs) and hardware spec (flops , HBM capacity) are unclear

**Top Reasons To Accept The Paper:**

None

**Top Reasons To Reject The Paper:**

None